# BCG in Bladder Cancer Immunotherapy

**DOI:** 10.3390/cancers14133073

**Published:** 2022-06-23

**Authors:** Song Jiang, Gil Redelman-Sidi

**Affiliations:** 1Urology Service, Memorial Sloan Kettering Cancer Center, New York, NY 10065, USA; jiangs1@mskcc.org; 2Infectious Diseases Service, Memorial Sloan Kettering Cancer Center, New York, NY 10065, USA

**Keywords:** BCG, bladder, immunotherapy

## Abstract

**Simple Summary:**

Bacillus Calmette–Guérin (BCG), a live attenuated strain of *Mycobacterium bovis*, is the most successful microbial immunotherapy of cancer in current use. Intravesical treatment with BCG is recommended for most patients with high-risk non-muscle invasive bladder cancer (NMIBC). In patients with NMIBC, treatment with BCG is associated with reduced risk of bladder cancer recurrence and progression compared to transurethral resection alone. Here, we summarize current data regarding the mechanism of BCG and review indications and recommended practice for BCG treatment of bladder cancer.

**Abstract:**

BCG is a live attenuated strain of *Mycobacterium bovis* that is primarily used as a vaccine against tuberculosis. In the past four decades, BCG has also been used for the treatment of non-muscle invasive bladder cancer (NMIBC). In patients with NMIBC, BCG reduces the risk of tumor recurrence and decreases the likelihood of progression to more invasive disease. Despite the long-term clinical experience with BCG, its mechanism of action is still being elucidated. Data from animal models and from human studies suggests that BCG activates both the innate and adaptive arms of the immune system eventually leading to tumor destruction. Herein, we review the current data regarding the mechanism of BCG and summarize the evidence for its clinical efficacy and recommended indications and clinical practice.

## 1. Introduction

Bacillus Calmette–Guérin (BCG) was first developed at the Pasteur Institute in 1921 by Albert Calmette and Camille Guérin by culturing *Mycobacterium bovis* in bile potato medium for over a decade. Worldwide, BCG is primarily used as vaccine against tuberculosis in children, in whom BCG vaccination results in modest protection from tuberculosis, particularly from its severe forms [1,2]. Beyond its use as a vaccine against tuberculosis, BCG is also the mainstay of the treatment of non-muscle-invasive bladder cancer (NMIBC). In its use for the treatment of bladder cancer, BCG is inarguably the most successful example, to date, of microbial treatment of cancer.

Microbial treatment of cancer was pioneered by the surgeon William B. Coley, who, in the late 19th century, employed intra-tumoral injections of S. pyogenes and Serratia marcescens to treat patients with cancer [3,4]. These mixtures, later termed Coley’s Toxins, are considered the first immunotherapy of cancer [5]. Shortly after development of the BCG vaccine, data emerged suggesting an inverse correlation between tuberculosis and cancer [6]. These observations led to studies, led by the immunologist Lloyd Old, demonstrating resistance to tumor implantation in mice infected with BCG [7], resulting in the discovery of tumor necrosis factor (TNF)-α [8]. Studies such as these sparked interest in the use of BCG to treat cancer. Subsequent clinical studies tested the utility of BCG as therapy for various cancers, including acute lymphoblastic leukemia and melanoma [9,10], eventually leading to the first study, in 1976, in which intravesical BCG was used to treat patients with NMIBC [11].

## 2. Mechanism of BCG

The mechanism of BCG therapy of bladder cancer is still not completely understood. Many mechanisms have been proposed, including direct interaction of BCG with urothelial and bladder cancer cells, activation of an innate immune response, and initiation of BCG-specific and tumor-specific T cell immunity (Figure 1). It is likely that the mechanism of BCG requires a combination of several of these factors.

### 2.1. Interactions between BCG and Urothelial Cells

Once it has been instilled into the bladder, the initial interaction between BCG and the recipient is contact between bacteria and the urothelium, including bladder cancer cells themselves. BCG can attach to urothelial cells, possibly facilitating downstream immune responses. The proposed mechanism of BCG’s attachment to the urothelium is through association of mycobacterial fibronectin attachment proteins (FAPs) to host fibronectin which, in turn, attaches to urothelial cells through integrin α5β1 [12,13,14]. One study demonstrated that blocking of BCG attachment to the urothelium via fibronectin abrogated the antitumor effect of BCG [15]; however, this finding was not reproduced in other studies [16,17].

Another focus of research has been the uptake and internalization of BCG by bladder cancer cells. Internalized BCG can be identified within the urothelial cells of patients receiving intravesical BCG [18]. In vitro, bladder cancer cell lines can internalize BCG via macropinocytosis, a non-specific mechanism for the uptake of large particles that depends on the activation of the serine/threonine kinase P21 activated kinase (Pak1) by certain tumor aberrations, including activating mutations of Ras and inactivating mutations of phosphatase and tensin homolog (PTEN) [19,20]. Bladder cancer cells that have internalized BCG secrete various immune-activating effectors, including interleukin (IL)-6, IL-8, granulocyte-macrophage colony-stimulating factor (GM-CSF) and tumor necrosis factor (TNF)-α [21,22,23,24], hinting at a possible mechanism in which internalization of BCG by bladder cells leads to recruitment or activation of immune cells. A few studies have suggested that bladder cancer cells could initiate an immune response by presenting antigen to CD4 T cells [25,26,27,28,29]; however, the ability to effectively present antigen and activate CD4 T cells is generally limited to professional antigen presenting cells.

Despite the in vitro data cited above, there is no conclusive in vivo data demonstrating that internalization of BCG by urothelial cells is required for the efficacy of BCG. A recent manuscript showed indirect evidence for this mechanism, demonstrating that BCG expressing the mannose-binding protein FimH had improved ability to adhere to urothelial cells and be internalized by them and also induced a more potent anti-tumor response as compared to wild type BCG. However, whether the improved efficacy of the FimH strain was through increased uptake by urothelial cells or via another mechanism was not shown [30].

Finally, BCG can exert direct cytotoxic effects on bladder cancer cells, inhibiting their growth and even killing them [31,32,33]. However, this phenomenon has primarily been demonstrated in vitro using high ratios of BCG to bladder cancer cells. Whether this mechanism is relevant in vivo remains uncertain.

### 2.2. Role of Innate Immunity

#### 2.2.1. Pattern Recognition Receptors

Activation of an innate immune response is initiated by binding of pathogen-associated molecular patterns (PAMPs) to host pattern recognition receptors (PRRs). BCG produces several PAMPs that can be recognized by host PRRs. These include the Toll-like Receptors (TLRs) TLR2, TLR4, and TLR9, all of which induce the MyD88 signaling pathway regulating the production of proinflammatory cytokines [34]. TLR2 is expressed on many innate immune cells, including monocytes and polymorphonuclear cells (PMNs), but also on B and T cells, and can be activated by several lipid components of mycobacteria [35,36,37,38]. TLR4 is expressed by myeloid cells and can bind mycobacterial heat-shock proteins, and TLR9 is expressed by dendritic cells, macrophages, and natural killer (NK) cells and can be activated by mycobacterial DNA [39]. Interestingly, in a mouse subcutaneous model of BCG treatment, MyD88-deficient mice had an abrogated response to BCG, supporting a role for TLR signaling in the efficacy of BCG [40].

Beyond TLRs, other PRRs that can react to BCG PAMPs include the complement receptors [41], the NOD-Like Receptor NOD2 [42], the C-type lectin-like receptor dectin-1 [43], and the dendritic cell adhesion molecule DC-SIGN [44].

#### 2.2.2. Macrophages

As compared to untreated patients, increased numbers of macrophages are found in the urine and bladder wall of BCG-treated patients [45,46,47] and their cytotoxic activity against bladder cancer cell lines is increased [48,49]. Interestingly, higher numbers of CD68+ CD163+ macrophages, commonly referred to as “M2” macrophages, are predictive of cancer recurrence after BCG therapy, suggesting that these macrophages may suppress the BCG-induced immune response [50,51,52].

#### 2.2.3. Polymorphonuclear Cells

Polymorphonuclear cells (PMNs) are the predominant cell type in the urine following BCG instillation [45]. In vitro, PMNs can kill BCG-infected bladder cancer cells [53]. In a mouse orthotopic bladder cancer model, depletion of PMNs is sufficient to abrogate the therapeutic effect of BCG, supporting an important role for PMNs in BCG efficacy [54].

#### 2.2.4. Dendritic Cells

Dendritic cells (DCs) are crucial for activation of T cells and have been hypothesized to play a role in BCGs mechanism. DCs can be found in the urine of BCG-treated patients with NMIBC [55] and BCG-pulsed dendritic cells can activate cytotoxic killing of cancer cells by natural killer T (NKT) cells and γδT cells [56,57]. In models of systemic BCG infection, basic leucine zipper transcription factor ATF-like 3 (BATF3)-dependent dendritic cells are required for Th1 responses to BCG [58]. Despite this supportive data, it has not been conclusively demonstrated that DCs are required for BCG immunotherapy of bladder cancer.

#### 2.2.5. Natural Killer Cells

NK cells are members of the rapidly expanding family of innate lymphoid cells (ILCs) [59]. NK cells demonstrate cytotoxicity against BCG-infected bladder cancer cells in vitro [60], and can be demonstrated in the immune infiltrate of bladder tumors in mice [61]. In mouse models of bladder cancer, depletion with an antibody targeting NK1.1 abrogates the survival benefit conferred by BCG, indicating that NK cells or NKT cells are required for the efficacy of BCG [61].

#### 2.2.6. Trained Immunity

The innate arm of the immune system has traditionally been viewed as lacking the capacity to form immune memory. In the past decade, our understanding of the innate immune system has evolved. It is now recognized that, upon stimulation, innate immune cells undergo epigenetic changes leading to an alteration in their capacity to react to restimulation with the same signal. This phenomenon has been termed **trained immunity** [62]. Moreover, these epigenetic changes alter not only responses to restimulation with the same signal but also to unrelated stimuli, a process that has been termed **heterologous immunity** [62]. Much of the evidence for trained immunity and heterologous immunity was derived from studies on the use of BCG as a childhood vaccine against tuberculosis. Vaccination with BCG protects not only from tuberculosis, its intended target, but also from unrelated infections, including respiratory viruses, in a process that depends on epigenetic changes in innate immune cells [63,64,65].

Trained immunity has also been proposed to play a role in the antitumor effect of BCG in bladder cancer [66]. Monocytes isolated from patients with bladder cancer after intravesical BCG demonstrated enhanced cytokine responses to ex vivo stimulation with lipopolysaccharide (LPS) compared to monocytes from the same patients before BCG treatment was initiated, suggesting that these monocytes had undergone immune training [67]. Furthermore, increased secretion of IL-12 by monocytes stimulated ex vivo with LPS after completion of five weekly instillations of BCG as compared to before BCG initiation was found to correlate with favorable clinical responses [68]. Further studies are ongoing to clarify the role of trained immunity in BCG treatment of bladder cancer.

### 2.3. Role of Adaptive Immunity

#### 2.3.1. T Cells

Given their critical role in immunity to mycobacterial infection, T cells have been a major focus of investigations into the mechanism of BCG treatment of bladder cancer. In animal models of bladder cancer, BCG is ineffective in the absence of T cells [69,70,71]. Both CD4 and CD8 T cells are required, and depletion of either subset results in loss of efficacy of BCG [70,71]. In humans, T cells, particularly of the CD4 lineage, are found in the urine of BCG-treated patients and can also be found infiltrating the bladder mucosa for months after BCG therapy [45,46,72].

BCG can shift the urinary cytokine milieu from T helper 2 (Th2) toward a T helper 1(Th1) profile, characterized by elevated levels of cytokines such as IL-2, IL-12 and interferon gamma (IFN-γ), and it has been proposed that this shift is required for a favorable clinical response to BCG [73,74]. Supporting this hypothesis, IL-12 and IFN-γ knockout mice fail to respond to BCG, while IL-10 knockout mice have improved outcomes after BCG treatment [75]. In humans, post-BCG urinary levels of the cytokines IL-2, IL-6, IL-8, IL-18, IL-1ra, TRAIL, IFN-γ, IL-12[p70], and tumor necrosis factor alpha, many of which are associated with a Th1 response, are predictive of clinical response [76].

#### 2.3.2. BCG-Specific versus Tumor-Specific Immunity

The antigenic target of the BCG-induced T cell immune response that determines clinical efficacy has been debated. One option is that T cell response is primarily aimed at BCG antigens. Supporting this, animal data demonstrates that subcutaneous vaccination with BCG prior to intravesical treatment results in increased infiltration of T cells into the bladder with improved elimination of bladder tumors [77]. In the same study, patients with a positive purified protein derivative (PPD) skin test were more likely to have a favorable clinical response to BCG, adding support to a mechanism in which mycobacterial immunity determines BCG response [77]. These results, however, contrast with results from a more recent study showing no survival benefit from transfer of BCG-specific T cells to bladder tumor-bearing mice [71].

A second option is that the BCG-induced T cell response is primarily aimed at tumor antigens. In mice, successful treatment of bladder tumors with intravesical BCG results in long term tumor-specific immunity that is primarily dependent on CD4 T cells. Adoptive transfer of T cells collected from mice cured of bladder cancer by BCG to tumor-bearing, treatment-naïve mice results in tumor rejection and survival benefit. In contrast, no survival benefit is gained if the T cells are transferred from mice who were treated with BCG in the absence of a bladder tumor. These results suggest that tumor-specific T cells are present in BCG-treated mice and that they play a role in the antitumor activity of BCG [71]. Preliminary human data also supports this option. CD4 T cells that recognize an MHC class II-restricted tumor neoantigen can be identified in BCG-treated patients with NMIBC [78]. Furthermore, one study, published in abstract form, found a relationship between higher tumor mutational burden and favorable BCG response, supporting a potential role for tumor neoantigens in BCG response [79].

A recent study confirmed that BCG induces T cell dependent tumor-specific immunity and suggested that BCG-specific γδT cells are required for induction of tumor-specific immunity mediated by conventional αβT. In this study, enhancement of γδT cell responses by treating mice with the mTOR inhibitor rapamycin enhanced the efficacy of BCG [80], a finding that has prompted clinical studies of this combination [81].

As opposed to T cells, there is scant evidence for the role of B cells and a humoral immune response in the mechanism of BCG.

### 2.4. Immune-Active Components of BCG

A potential disadvantage of the use of live BCG for bladder cancer treatment is that it can result in infectious complications in some patients (see Safety and Potential Complications below). For this reason, investigators have attempted to determine the immune-active components of BCG that could achieve its antitumor efficacy without the risk of infectious complications. Potential candidates include the PRR activating components of BCG described above, lipid components such as trehalose dimycolate (cord factor) [82], immunogenic mycobacterial proteins such as Ag85b, mycobacterial components recognized by the different varieties of γδT cells [83], and the BCG-cell wall skeleton [84].

## 3. Clinical Efficacy

### 3.1. Recurrence-Free Survival

Current management of NMIBC involves initial complete resection accomplished via transurethral resection of bladder tumor (TURBT) with postoperative surveillance by cystoscopy. Without adjunctive post-resection therapy, cancer recurs in 50–70% of patients with NMIBC, particularly those with higher risk tumors, even with apparent complete resection of bladder tumors. Adjuvant treatment with intravesical BCG is the most effective therapy in reducing the risk of recurrence in patients with intermediate and high-risk NMIBC [85].

An early study demonstrated a 5-year recurrence free survival (RFS) of 80% in those treated with intravesical BCG in combination with percutaneous vaccination, compared to only 48% in those who did not receive BCG [86]. In this study, positive PPD skin test predicted response to intravesical BCG [86]. A similar benefit in RFS among BCG-treated patients was seen in a concurrent randomized controlled study where patients underwent TURBT alone or followed by intravesical BCG given in six weekly doses [87].

The benefit of adjuvant intravesical BCG therapy in reducing tumor recurrence was confirmed by several subsequent studies [88,89,90,91] (Table 1). A pooled analysis and Cochrane review showed a significant advantage of BCG in reducing the likelihood of disease recurrence at 12 months compared to TURBT alone (OR 0.30, 95% CI 0.21–0.43) [92]. Particularly with high-risk disease, BCG-treated patients had better RFS than those treated with intravesical chemotherapeutic agents, including thiotepa, doxorubicin, epirubicin and mitomycin [93,94,95,96]. As a result of these landmark studies, BCG was established as the standard of care adjuvant therapy for the treatment of NMIBC. Contemporary retrospective studies, which incorporate more standardized clinical practices including the institution of re-staging TUR as well as maintenance BCG therapy, continue to support the use of adjuvant intravesical BCG therapy as a benchmark for the management of high-risk NMIBC [97,98,99] (Table 1).

### 3.2. Progression-Free Survival

Approximately 50% of patients with NMIBC who are treated with TURBT without additional treatment experience tumor progression to muscle-invasive or metastatic disease. Evidence suggests that BCG reduces this risk of tumor progression. In an early randomized trial of 86 patients, Herr and colleagues demonstrated that treatment with intravesical BCG after TURBT granted a benefit to progression-free survival (PFS) compared to TURBT alone [100]. A subsequent analysis of the same cohort after 10 years of follow-up demonstrated 62% PFS for BCG-treated patients, compared to 37% for those who received TURBT alone [101]. Similar benefits in PFS were reported in other studies, including in contemporary large retrospective series [89,90,97,98,99,102] (Table 2). Consistent with these findings, a meta-analysis incorporating 24 trials with a median follow up of 2.5 years demonstrated a 27% reduction in the odds of progression for patients receiving BCG, and also highlighted the importance of maintenance BCG treatment (see below) in improving PFS [103]. Furthermore, when compared to intravesical chemotherapy, investigators demonstrated reduced rates of metastatic disease and better overall and disease-specific survival in patients treated with BCG [93].

The impact of BCG in improving overall survival remains inadequately assessed, but population-scale retrospective analysis of Surveillance, Epidemiology, and End Results (SEER)-Medicare data concluded that BCG significantly reduced overall mortality (HR 0.87), and bladder cancer deaths in patients with high-grade NMIBC. This study also underlined the fact that only 22% of eligible patients receive adequate BCG therapy, with only a subset of these patients receiving guideline-recommended maintenance courses [104].

A potentially important issue with assessing the efficacy of BCG in reducing the risk of recurrence and improving progression-free survival is the timing of BCG after tumor resection. A European group found that initiation of BCG within 6 weeks of TURBT was associated with a lower rate of cancer recurrence than when BCG treatment was initiated at a later time point [105,106]. Conversely, in an American study of 518 patients time to initiation of BCG after TURBT was not significantly associated with therapeutic response [107].

### 3.3. Role of Maintenance BCG Treatment

BCG may be administered across two discrete phases—an initial induction phase that typically consists of six weekly administrations of BCG, and a subsequent maintenance therapy that consists of less frequent administration of BCG. As mentioned above, several trials have suggested that the improved PFS associated with BCG relies on adequate maintenance therapy. A full 6-week induction course of BCG results in a RFS of 25–74%, while additional maintenance therapy can result in an approximately 28% reduction of tumor recurrence compared to induction alone [108,109].

The current 3-week maintenance schedule associated with the Southwest Oncology Group (SWOG) 8507 trial demonstrated that, compared to standard induction therapy, maintenance BCG treatment resulted in median RFS time that was twice as long in the 3-week maintenance arm with significantly longer PFS. In the subpopulation of patients with carcinoma in situ (CIS), complete response at 6 months in patients randomized to receive the initial 6-week BCG treatment was only 69% in comparison with 84% in patients randomized to receive the additional 3-week maintenance treatment. In those who received maintenance therapy, complete response increased from 55% at 3 months to 84% at 6 months (64% complete response in patients with treatment failure at 3 months). Even without additional BCG, 26% of patients with residual disease at 3 months in the induction arm went on to have complete response by 6 months [109].

Notably, randomized controlled trials of quarterly, monthly, and repeated 6-week BCG instillations have failed to demonstrate improved outcomes of maintenance BCG treatment [110,111,112]. The lack of a dose response associated with maintenance therapy led some experts to question the value of maintenance BCG treatment and to abandon its use. Herr and colleagues demonstrated, in a cohort of 816 high-risk NMIBC patients who had initial response to induction BCG and who did not receive subsequent maintenance therapy, outcomes comparable to those of patients from trials in which maintenance therapy was used. Thirty-two percent of the patients required another course of BCG therapy for relapsing disease. The conclusions of this study were that, in a setting of appropriate restaging and complete resection of high-risk disease, maintenance BCG treatment may not be required [97].

In a metanalysis performed by Chen and colleagues, maintenance BCG treatment resulted in a relative risk reduction of approximately 21% in tumor recurrence and 33% reduction in disease progression compared to induction treatment alone [113]. Sylvester et al. further demonstrated in the European Organization for Research and Treatment of Cancer (EORTC) 30911 trial—which compared 3-week maintenance BCG treatment with 3-week maintenance intravesical epirubicin chemotherapy—that in patients with intermediate and high-risk stage Ta and T1 disease, intravesical BCG was superior to intravesical epirubicin maintenance therapy. This was the case not only for time to first recurrence but also for time to distant metastases, overall survival, and disease-specific survival [114]. For these reasons, maintenance BCG therapy remains as a part of guideline recommendations for the treatment of intermediate and high-risk NMIBC.

## 4. Clinical Indications and Recommended Treatment Schedule

### 4.1. Risk Categories Associated with NMIBC

Histologic grading of papillary urothelial carcinoma is of prognostic value in NMIBC. Two iterations of classifications from the World Health Organization (WHO) from 1973 and 2004/2016 are utilized in current guidelines in NMIBC. Both classification systems use a three-point classification, with WHO 1973 using numerical grades 1, 2, and 3, and WHO 2004/2016 using the three classes of papillary urothelial neoplasm of low malignant potential (PUNLMP), low-grade, and high-grade. In a recent study, investigators determined that a combination of both classification systems proved to be superior to either classification system alone, as it segregates the large group of G2 patients from WHO 1973 into two subgroups (low-grade and high-grade) with distinct prognoses [115].

Integrating these histologic classifications, tumor stage, associated presence of CIS, and patient factors such as gender, age, prior recurrence status, tumor size, and tumor multiplicity, the Spanish Urology Association for Oncological Treatment (CUETO) and the European Organization for Research and Treatment of Cancer (EORTC) developed independent models to predict the risk of recurrence and progression in NMIBC patients who are treated with adjuvant intravesical therapy [114,116]. The EORTC found that prior recurrences and tumor multiplicity were the most important prognostic factors for disease recurrence, while stage and histologic grade were determinants of disease progression and disease-specific survival [114]. Low, intermediate, and high-risk NMIBC groups were developed using these prognostic models and serve as the basis for the nomograms for BCG-treated patients developed by both the American Urologic Association (AUA) as well as the European Association of Urology (EAU) [117,118].

### 4.2. Carcinoma In Situ

CIS represents a distinct subset of NMIBC that often portends worse prognosis secondary to the inherent difficulty of complete resection and also likely due to the divergent biology of this entity. Patients with tumors containing a CIS component represent a high-risk population, with inconsistent representation in clinical trials. Furthermore, CIS can occur as a primary, concurrent, or secondary disease, resulting in a heterogeneity of clinical outcome. Primary CIS has been reported in approximately 3% of all patients with bladder cancer. More commonly, it is found concurrently with T1 disease or with muscle-invasive disease (T2–T4) [119]. The natural history of CIS in patients treated with resection and/or fulguration alone has demonstrated progression to muscle-invasive disease in 54% of patients after 4 years [120]. A similar rate of 53% progression has been reported with 15 years of follow-up [121]. A recent study showed that older patients (age > 70 years) were more likely to present with CIS, and that in patients with pure CIS, older age was an independent predictor of poor outcome [122]. Additionally, diagnosis of CIS is complicated by significant interobserver variability of both the surgeon and the pathologist. Advanced imaging modalities as well as new urinary biomarkers continue to be an active area of pursuit to help improve detection [123].

In addition to its role in papillary disease, intravesical BCG is the generally accepted standard therapy for CIS. Although the early disease-free rates of BCG therapy for CIS are high, the treated natural history generally reflects high rates of recurrence and progression [124,125]. The SWOG 8507 trial comparing induction therapy alone with induction plus maintenance demonstrated not only the benefit of maintenance BCG treatment but also the potential of a delayed benefit of BCG in the treatment of CIS. This group advocated waiting for 6 months before assessing response to treatment with a repeat bladder biopsy [109]. However, long-term data on clinical outcomes of CIS after maintenance therapy remain lacking as most initial trials did not report outcomes with BCG specific to patients with CIS.

### 4.3. Current Treatment Schedule Guidelines

Current AUA and EAU guidelines on NMIBC recommend an induction course of BCG for high-risk disease. This is followed by one to three years of maintenance BCG treatment for BCG-responsive disease. For relapsed disease after BCG treatment, guideline recommendations for intermediate and high-risk categories disease include the institution of a second induction BCG course, citing evidence showing that approximately 50% of patients with recurrent NMIBC respond to a second induction course [126].

Induction BCG instillations are given according to the empirical weekly schedule for six weeks originally introduced by Morales and colleagues [11]. In a metanalysis by Bohle et al., it was shown that at least one year of maintenance BCG treatment was required to obtain superiority of BCG over intravesical mitomycin C for prevention of recurrence or progression based on a pooled analysis of 9 eligible clinical trials involving 2400 patients [127].

The EORTC conducted a subsequent multi-arm phase III randomized study of 1355 patients to identify optimal dosing schedule and population for maintenance BCG treatment. In this study, induction BCG given at full dose with three years of maintenance therapy (weekly instillations for three weeks at 3, 6, 12, 18, 24, 30 and 36 months) reduced the recurrence rate compared to one year of maintenance therapy in high-risk NMIBC patients, but not in intermediate-risk patients. There were no differences in progression or overall survival for either group [93].

Grimm et al. recently evaluated, in a prospective randomized controlled trial, the optimal number of induction instillations and frequency of maintenance instillations. A reduced number of instillations (three instillations in induction and two instillations at 3, 6 and 12 months) was inferior to the standard schedule (six instillations in induction and three instillations at 3, 6 and 12 months) in time to first recurrence [128].

A recent phase 2 prospective study evaluated the safety and efficacy of two sequential BCG induction courses (12 weekly treatments). Of 76 patients who initiated the trial, 63 (83%) completed the 12 instillations on schedule, and 62 of these had complete response at 6 months with none experiencing tumor progression and 1 patient experiencing a serious adverse event. Two-year recurrence-free survival in those with complete response was 85%, comparing favorably with published results for the standard induction course [129].

The past decade has seen international shortages of BCG, mainly related to difficulty in scaling up production in the face of fewer approved manufacturers and growing demand [130]. Given these shortages, instillation of a reduced dose of BCG has been proposed. The EORTC 30962 trial demonstrated similar toxicity between one-third dose and full dose as well as non-inferiority in the efficacy of the reduced dose [128]. Martinez-Pineiro and colleagues in association with the CUETO group also compared one-third dose to full-dose BCG and found no overall difference in efficacy. One-third of the standard dose of BCG might be the minimum effective dose for intermediate-risk tumors, as a further reduction to one-sixth dose resulted in a decrease in efficacy with no decrease in toxicity [131,132].

In addition to the difficulties related to the BCG shortages mentioned above, the COVID-19 pandemic has also adversely impacted the receipt of timely BCG treatment and appropriate administration of maintenance BCG therapy [133].

Currently, the standard of care remains a full-dose induction regimen given in six weekly courses followed by a three-week maintenance course. However, there is significant heterogeneity in clinical practice regarding the length and schedule of the maintenance course. The current AUA guidelines recommend that high-risk patients receive three years of maintenance BCG treatment after induction therapy, while EAU guidelines provide a range of one to three years for both intermediate and high-risk patients after their induction course [118,134].

## 5. Safety and Potential Complications

As compared to intravesical chemotherapy, BCG is more commonly associated with side effects. Nevertheless, serious side effects are rare and are usually treatable. The most common side effects of BCG are irritative urinary symptoms, such as dysuria, urinary frequency, and urinary urgency. Such symptoms are usually transient. Patients in whom such symptoms persist should be evaluated for the presence of bacterial cystitis, and, if confirmed, treated with appropriate antibiotics. Additionally, common are systemic toxicities such as fever, malaise, and skin rash. These can occur in up to 39% of patients but are also usually transient and require treatment cessation in fewer than 10% of cases [93,135]. BCG maintenance treatment is not associated with an increased risk of side effects compared to an induction course alone [135].

Additional complications are the result of the establishment of a persistent BCG infection. These include local spread of BCG resulting in cystitis, prostatitis, epididymo-orchitis, or, rarely, pyelonephritis. More infrequently, BCG can spread to distant organs, resulting in hepatitis, pneumonitis, mycotic aneurysms, or osteoarticular and soft tissue infection [136]. A very small fraction of treated patients develop life-threatening sepsis [136]. Overall, BCG infection occurs in as many as 1% of treated patients. Known risk factors for hematogenous spread of BCG include disruption of the urothelial barrier at the time of instillation due to traumatic urinary catheterization, early instillation of BCG after TURBT before adequate healing has occurred, or concurrent urinary tract infection, all of which represent contraindications to intravesical BCG administration [126]. BCG infection is treated with antimycobacterial antibiotics, usually under the supervision of an infectious diseases specialist.

## 6. Prognostic Markers

Prognostic markers determining response to BCG are beyond the scope of this review. For a recent review comprehensively assessing the factors associated with clinical response after BCG treatment please see Li and Kamat 2020 [137].

## 7. Conclusions

BCG is the treatment of choice for patients with intermediate and high-risk NMIBC. In these patients, BCG reduces the risk of cancer recurrence and progression and likely improves overall survival. Our understanding of the mechanism of BCG is evolving, with current data suggesting a complex interplay between innate immune cells, T cells, and cancer cells. As we elucidate the mechanism of BCG, our understanding of the factors determining clinical response continues to improve. These insights will allow us to better tailor the choice of initial NMIBC treatment to the individual patient and improve the BCG treatment of bladder cancer, either by design of more effective strains of BCG or through synergistic combination of BCG with other forms of treatment.

## Figures and Tables

**Figure 1 cancers-14-03073-f001:**
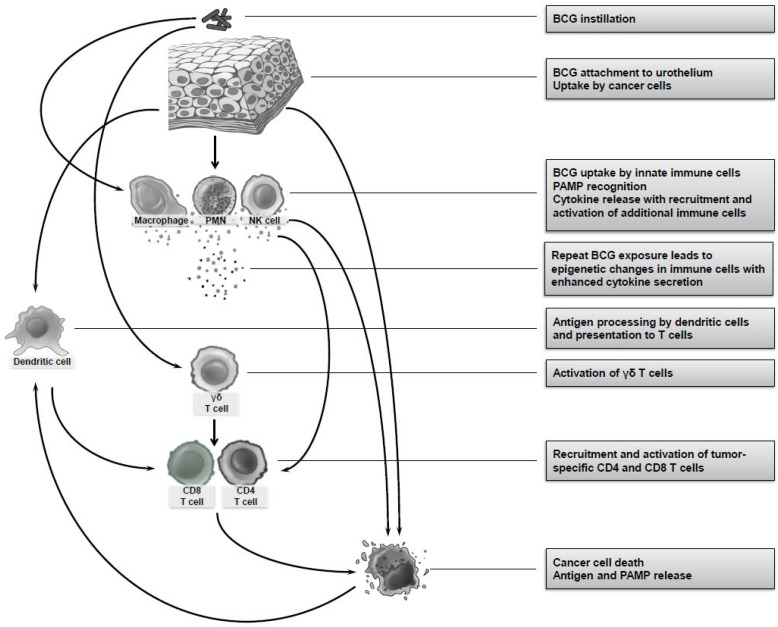
Proposed mechanisms of BCG therapy of bladder cancer.

**Table 1 cancers-14-03073-t001:** Clinical trials assessing recurrence-free survival after BCG treatment.

Author, Year	*n*	BCG Dose	BCG Schedule	1-Year Recurrence Events/Patients or %	Relative Risk of 1-Year Recurrence
Lamm, 1985 [86]	60	120 mg intravesical Pasteur + 5 mg percutaneous	Weekly × 6 weeks	3/6 BCG 4/6 no BCG	0.75
Pinsky, 1985 [87]	88	120 mg intravesical Armand Frappier + 5 × 10^7^ CFU percutaneous	Weekly × 6 weeks	27/41 BCG 39/43 no BCG	0.73
Yamamoto, 1990 [88]	44	80 mg Tokyo	Weekly × 6 weeks, every 2 weeks × 12 weeks, every month × 20 months	4/20 BCG 13/20 no BCG	0.31
Pagano, 1991 [89]	189	75 mg intravesical Pasteur	Weekly × 6 weeks, monthly × 1 year, then quarterly × 1 year	6/43 BCG 31/39 no BCG	0.18
Melekos, 1990 [90]	100	150 mg intravesical Pasteur	Weekly × 6 weeks, every 3 months × 2 years	11/67 BCG 14/33 no BCG	0.38
Krege, 1996 [91]	337	120 mg Connaught	Weekly × 6 weeks, monthly × 4 months	28/98 BCG 43/116 no BCG	0.78
Herr, 2011 [97]	816	50 mg TICE	Weekly × 6, no maintenance	27%	-
Sfakianos, 2014 [98]	1021	50 mg TICE	Weekly × 6, no maintenance	28%	-
Matulay, 2021 [99]	542	50 mg TICE	Weekly × ≥5 induction plus ≥2 maintenance	8%	-

**Table 2 cancers-14-03073-t002:** Clinical trials assessing progression-free survival after BCG treatment.

Author, Year	*n*	BCG Dose	BCG Schedule	Progression Events/Patients or %	Relative Risk of Progression
Herr, 1988 [100]	88	120 mg intravesical Armand Frappier + percutaneous 5 × 10^7^ CFU	Weekly × 6 weeks	23/43 BCG 41/43 no BCG	0.56
Melekos, 1993 [102]	94	NA	Weekly × 6 weeks, plus weekly × 4 weeks	4/62 BCG 7/32 no BCG	0.29
Melekos, 1990 [90]	100	150 mg intravesical Pasteur	Weekly × 6 weeks, every 3 months × 2 years	7/67 BCG 13/33 no BCG	0.27
Pagano, 1991 [89]	189	75 mg intravesical Pasteur	Weekly × 6 weeks, monthly × 1 year, then quarterly × 1 year	3/70 BCG 11/63 no BCG	0.25
Herr, 2011 [97]	816	50 mg TICE	Weekly × 6, no maintenance; 2nd weekly × 6 course with tumor relapse	11%	-
Sfakianos, 2014 [98]	1021	50 mg TICE	Weekly × 6, no maintenance	18%	-
Matulay, 2021 [99]	542	50 mg TICE	Weekly × ≥5 induction plus ≥2 maintenance	9%	-

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
