# Peer review of "BCG in Bladder Cancer Immunotherapy"

_cancers, 2022, doi:10.3390/cancers14133073_

Round 1

Reviewer 1 Report

The authors answered all comments and suggestions.

Author Response

Thank you.

Reviewer 2 Report

The authors have considered all of the reviewer's criticisms and revised the manuscript accordingly.

Author Response

Thank you.

Reviewer 3 Report

Thank you for your corrections.

Only the sentence in lines 413-414 still needs some modification, e. g.  “The EORTC 30962 trial demonstrated similar toxicity between one-third dose and full dose but did not demonstrate inferior efficacy of the reduced dose.” (I am sorry for my previous recommendation that was not quite correct)

Author Response

Thank you.

Editorial team and editors confirm this can be updated during proofreading: "Only the sentence in lines 413-414 still needs some modification, e. g.  “The EORTC 30962 trial demonstrated similar toxicity between one-third dose and full dose but did not demonstrate inferior efficacy of the reduced dose.”

Reviewer 4 Report

The manuscript could be improved by adding original figures and updating the data of the tables. The new figure is the same that another figure previously published in two different journals:

https://www.nature.com/articles/nrurol.2014.15

https://onlinelibrary.wiley.com/doi/10.1111/iju.13410

The authors have added a couple of studies but they are not representative of all the work done in this field.

Author Response

Editor 2 comments: 

the authors revised the figure compared to the original manuscript.

No further changes are needed.

Accept as it is.

Editor 1 comments: Accept in current form

Author response: 

We thank the reviewer for their suggestions. Table 1 and 2 were updated with more recent studies. A figure depicting possible mechanisms of BCG was added.

The figure is significantly changed and takes into account newer insights into the mechanism of BCG, including the role of gamma delta T cells and the possible role of trained immunity

This manuscript is a resubmission of an earlier submission. The following is a list of the peer review reports and author responses from that submission.

Round 1

Reviewer 1 Report

The current study aims to summarize current data regarding the mechanism of BCG, and review indications and recommended practice for BCG treatment of bladder cancer.

The authors should be congratulated for the work and for addressing an important topic. However, several points warrant mentions:

Comments:

  1. Since there aren’t patients information about the time of starting BCG therapy, this can affect the results. Authors should be clearer about the how long it takes from primary diagnosis to the first BCG therapy.
  2. Redelman-Sidi G et. al (https://pubmed.ncbi.nlm.nih.gov/24492433/) observed that there is also another mechanism of BCG therapy, in particular they describe how the exposure of bladder cancer cells to BCG can result in decreased proliferation and cell cycle arrest, and internalization of BCG by bladder cancer cells can induce cell death.
  3. I would suggest to add tables to resume the timing of maintenance BCG for BCG-responsive disease.
  4. Authors should discuss about the appropriate frequency of surveillance based on stage/grade of bladder cancer after BCG therapy.
  5. I believe that authors should include more information also in view of current literature. This interesting paper deserve to read about:

    -https://pubmed.ncbi.nlm.nih.gov/34771440/ DOI: 10.3390/cancers13215276

          -https://pubmed.ncbi.nlm.nih.gov/35033480/; DOI: 10.1016/j.clgc.2021.12.005

The authors should be congratulated for the work and for addressing an important topic.

I believe that the study has sufficient merit to be considered for publication on Cancers, although major revisions are required.

Author Response

  1. Since there aren’t patients information about the time of starting BCG therapy, this can affect the results. Authors should be clearer about the how long it takes from primary diagnosis to the first BCG therapy.

We thank the reviewer for this comment. Based on the recommendation we added a paragraph about the potential importance of timing of BCG after the initial diagnosis.

  1. Redelman-Sidi G et. al (https://pubmed.ncbi.nlm.nih.gov/24492433/) observed that there is also another mechanism of BCG therapy, in particular they describe how the exposure of bladder cancer cells to BCG can result in decreased proliferation and cell cycle arrest, and internalization of BCG by bladder cancer cells can induce cell death.

We thank the reviewer for this comment. The manuscript mentions the direct effect of BCG on bladder cancer cells with the caveat that most of the data regarding this mechanism is from in vitro studies.

  1. I would suggest to add tables to resume the timing of maintenance BCG for BCG-responsive disease.

We thank the reviewer for this comment. We have detailed the recommendations regarding the maintenance in the text and feel that a separate table is not required in addition to this.

  1. Authors should discuss about the appropriate frequency of surveillance based on stage/grade of bladder cancer after BCG therapy.

We thank the reviewer for this comment. We believe that discussion about the frequency of surveillance is beyond the scope of this review.

  1. I believe that authors should include more information also in view of current literature. This interesting paper deserve to read about:

    -https://pubmed.ncbi.nlm.nih.gov/34771440/ DOI: 10.3390/cancers13215276

          -https://pubmed.ncbi.nlm.nih.gov/35033480/; DOI: 10.1016/j.clgc.2021.12.005

We thank the reviewer for this comment. Both of these papers were added to the review.

Reviewer 2 Report

In the submitted manuscript, Jian and Redelman-Sidi provide a nearly comprehensive and up-to-date review on the use of BCG in the treatment of NMIBC. They discuss in detail the principal mechanisms of action of BCG and various immunological aspects of this activity. They highlight in a very understandable and concise manner the role of different immune cell types in innate and trained immunity. Furthermore, a detailed overview of the clinical therapeutic efficacy of BCG with respect to different endpoints, the benefit of maintenance therapy as well as different NMIBC risk categories is given.

Only a minor revision seems necessary. The authors should go into further detail on the relevant and potentially effective components of BCG, which in the end determine the complex effects of the therapeutic agent.

In line 192 on page 5 a year number seems to be missing as well as details on author contribution, funding and conflicts of interest on page 9.

Author Response

In the submitted manuscript, Jian and Redelman-Sidi provide a nearly comprehensive and up-to-date review on the use of BCG in the treatment of NMIBC. They discuss in detail the principal mechanisms of action of BCG and various immunological aspects of this activity. They highlight in a very understandable and concise manner the role of different immune cell types in innate and trained immunity. Furthermore, a detailed overview of the clinical therapeutic efficacy of BCG with respect to different endpoints, the benefit of maintenance therapy as well as different NMIBC risk categories is given.

Only a minor revision seems necessary. The authors should go into further detail on the relevant and potentially effective components of BCG, which in the end determine the complex effects of the therapeutic agent.

We thank the reviewer for this suggestion. A section on potential active components of BCG was added.

In line 192 on page 5 a year number seems to be missing as well as details on author contribution, funding and conflicts of interest on page 9.

We thank the reviewer for this comment. We have corrected line 192 and added the requested details to the manuscript.

Reviewer 3 Report

This review paper describes the possible mechanisms of therapy of non-muscle invasive bladder cancer (NMIBC) with Bacillus Calmette Guerin (BCG). The clinical indications and recommended practice for BCG treatment of NMIBC are also reviewed. Despite the long-term use of BCG for bladder cancer therapy, mechanisms leading to an antitumor effect of intravesical administration of BCG are still a topical subject that is covered with the up-to-date data in the manuscript.

The Simple Summary was not provided. As the Abstract is a bit short (82 words), it could be used as the Simple Summary and a more detailed Abstract could be prepared.

A figure (e. g. possible mechanisms of the BCG antitumor effect) would freshen the review up.

Some abbreviations were not introduced (e. g. IL, GM-CSF, PAK1, PTEN, BATF3). The abbreviation “OS” was introduced but was not used thereafter.

Numbering of chapters is a bit confusing. I recommend changing 3. Trained immunity into 2.2.6 and 3.1. Role of adaptive immunity into 2.3. with corresponding modifications bellow.

Author Contributions, Funding, and Conflicts of Interest are missing.

The format of the references needs to be unified.

Minor comments:

Lines 62-64: I am afraid that bladder cancer cells cannot “initiate an immune response” as they are not professional antigen presenting cells. Can you specify T cells that are activated?

Line 81: Please correct “can by activated”.

Lines 87-88: Macrophages are also present in the urine and bladder wall of nontreated patients, but their infiltration is increased and cytotoxicity enhanced after BCG treatment, which could be mentioned in the manuscript.

Line 97: Can you add the citation?

Lines 101-102: Rather DCs, not cancer cells, were infected with BCG.

Lines 109-110: NK cells can also infiltrate bladder tumors not treated with BCG.

Lines 110-112: Only one bladder cancer model was used. The antibody against NK1.1, which was used in the study, can also deplete NKT cells.

Lines 129-130: Please correct “… BCG was initiated” (e. g. “BCG treatment was initiated”). Similar in line 218 (“maintenance BCG”), titles of Tables (“after BCG”), 8. Conclusions, and elsewhere.

Lines 145-150: Introduction of the abbreviation “IFN-g” is not properly located.

Line 148: Please correct “… after BCG” (e. g. “… after BCG treatment”).

Line 156: Are you sure that the T cells were specific for BCG (was specificity tested in the study)?

Line 192: Please correct “An early study in demonstrated…”.

Lines 199 and 215: I recommend to use the order “ … subsequent/other studies [references] (Table x).“ because Tables do not contain only the subsequent/other studies.

Lines 293-294: Please be consistent: in situ vs. in-situ (see also et al vs. et al.). The CIS abbreviation was introduced above (line 242).

Line 296: Please correct “… an also likely…” and “…CIS-containing tumors …”.

Line 310: Please correct “… the treated natural history generally reflect …”.

Line 332: Please correct “… instillations 3, 6, 12, 18, 24, 30 and 36 months …”.

Line 341: Can you specify BCG shortages?

Lines 342-343: I recommend this modification: … demonstrated similar toxicity between one-third dose and non-inferior efficacy of the reduced dose (or: … but did not demonstrate inferior efficacy …).

Line 391: Please correct “… treatment of BCG…” (treatment with BCG, treatment of NMIBC?).

Author Response

his review paper describes the possible mechanisms of therapy of non-muscle invasive bladder cancer (NMIBC) with Bacillus Calmette Guerin (BCG). The clinical indications and recommended practice for BCG treatment of NMIBC are also reviewed. Despite the long-term use of BCG for bladder cancer therapy, mechanisms leading to an antitumor effect of intravesical administration of BCG are still a topical subject that is covered with the up-to-date data in the manuscript.

The Simple Summary was not provided. As the Abstract is a bit short (82 words), it could be used as the Simple Summary and a more detailed Abstract could be prepared.

We thank the reviewer for this comment. We have added a simple summary and have expanded the abstract.

A figure (e. g. possible mechanisms of the BCG antitumor effect) would freshen the review up.

We thank the reviewer for this suggestion. A figure depicting possible mechanisms of BCG was added.

Some abbreviations were not introduced (e. g. IL, GM-CSF, PAK1, PTEN, BATF3). The abbreviation “OS” was introduced but was not used thereafter. PAK1

Thank you for this comment. We have corrected the abbreviations in the manuscript as recommended.

Numbering of chapters is a bit confusing. I recommend changing 3. Trained immunity into 2.2.6 and 3.1. Role of adaptive immunity into 2.3. with corresponding modifications bellow.

The chapter numbers have been changed as recommended.

Author Contributions, Funding, and Conflicts of Interest are missing.

Author contributions, funding and conflicts of interest were added

The format of the references needs to be unified.

 Format of references was unified.

Minor comments:

Lines 62-64: I am afraid that bladder cancer cells cannot “initiate an immune response” as they are not professional antigen presenting cells. Can you specify T cells that are activated?

Thank you for the comment. While bladder cancer cells are not antigen presenting cells, several studies cited in the paragraph did find that some bladder cancer cell lines were able to directly activate CD4 T cells, though we agree that this has not been universally demonstrated in vitro and has never been shown to occur in vivo. Per the reviewer’s recommendation, we have added a subsequent sentence stating these limitations.

Line 81: Please correct “can by activated”.

Corrected.

Lines 87-88: Macrophages are also present in the urine and bladder wall of nontreated patients, but their infiltration is increased and cytotoxicity enhanced after BCG treatment, which could be mentioned in the manuscript.

Thank you for this comment. The sentence was corrected.

Line 97: Can you add the citation?

Thank you for the correction. The missing citation was added.

Lines 101-102: Rather DCs, not cancer cells, were infected with BCG.

We thank the reviewer for this comment. The sentence was corrected.

Lines 109-110: NK cells can also infiltrate bladder tumors not treated with BCG.

Thank you for this comment. The sentence was emended.

Lines 110-112: Only one bladder cancer model was used. The antibody against NK1.1, which was used in the study, can also deplete NKT cells.

We thank the reviewer for pointing this out. The sentence was corrected.

Lines 129-130: Please correct “… BCG was initiated” (e. g. “BCG treatment was initiated”). Similar in line 218 (“maintenance BCG”), titles of Tables (“after BCG”), 8. Conclusions, and elsewhere.

Thank you for this comment. This was corrected in the relevant locations.

Lines 145-150: Introduction of the abbreviation “IFN-g” is not properly located.

Thank you for the correction. This was emended.

Line 148: Please correct “… after BCG” (e. g. “… after BCG treatment”).

This was corrected in line 148 and in all other locations in the text.

Line 156: Are you sure that the T cells were specific for BCG (was specificity tested in the study)?

Thank you for this important point. We have corrected the sentence.

Line 192: Please correct “An early study in demonstrated…”.

We thank the reviewer. The sentence was corrected.

Lines 199 and 215: I recommend to use the order “ … subsequent/other studies [references] (Table x).“ because Tables do not contain only the subsequent/other studies.

Thank you for this comment. The sentences were corrected.

Lines 293-294: Please be consistent: in situ vs. in-situ (see also et al vs. et al.). The CIS abbreviation was introduced above (line 242).

Thank you. We have unified the formatting of in situ and et al. throughout the manuscript and corrected the introduction of the CIS abbreviation.

Line 296: Please correct “… an also likely…” and “…CIS-containing tumors …”.

Thank you. This was corrected.

Line 310: Please correct “… the treated natural history generally reflect …”.

Thank you. This was corrected.

Line 332: Please correct “… instillations 3, 6, 12, 18, 24, 30 and 36 months …”.

Thank you for the comment. The line was corrected.

Line 341: Can you specify BCG shortages?

Details on the BCG shortages were added.

Lines 342-343: I recommend this modification: … demonstrated similar toxicity between one-third dose and non-inferior efficacy of the reduced dose (or: … but did not demonstrate inferior efficacy …).

Line was corrected as suggested

Line 391: Please correct “… treatment of BCG…” (treatment with BCG, treatment of NMIBC?).

We thank the reviewer for their thorough review of the manuscript. The line was corrected.

Reviewer 4 Report

The authors revise the role of BCG on bladder cancer treatment. The provided information should be updated. For instance, recent studies can be included in Tables 1 and 2. The inclusion of some figures related to BCG mechanism can improve the manuscript.

Author Response

We thank the reviewer for their suggestions. Table 1 and 2 were updated with more recent studies. A figure depicting possible mechanisms of BCG was added.